# Screening out microRNAs and Their Molecular Pathways with a Potential Role in the Regulation of Parvovirus B19 Infection Through In Silico Analysis

**DOI:** 10.3390/ijms26115038

**Published:** 2025-05-23

**Authors:** Vívian de Almeida Salvado, Arthur Daniel Rocha Alves, Wagner Luis da Costa Nunes Pimentel Coelho, Mayla Abrahim Costa, Alexandro Guterres, Luciane Almeida Amado

**Affiliations:** 1Laboratory of Technological Development in Virology, Oswaldo Cruz Institute (IOC), FIOCRUZ, Rio de Janeiro 21040-360, Brazil; viviansalvado@id.uff.br (V.d.A.S.); guterres_rj@yahoo.com.br (A.G.); 2Competitive Intelligence Office, Institute of Immunobiological Technology (Bio-Manguinhos), FIOCRUZ, Rio de Janeiro 21040-360, Brazil; mayla.costa@bio.fiocruz.br

**Keywords:** Parvovirus B19, microRNAs, in silico, pathway, target gene

## Abstract

Parvovirus B19 (B19V) infection in healthy individuals is commonly asymptomatic or has non-specific symptoms, such as fever, headache, chills, myalgia, rash, and arthralgia. However, some groups of individuals, such as pregnant women, patients with hemolytic disorders, and immunocompromised individuals, may present severe forms of the infection, which may even lead to a negative outcome. To better understand what leads to this divergence of outcomes in different populational groups, this study sought to analyze the role of miRNAs in the pathogenesis of B19V infection. The miRNAs that potentially bind to the B19V transcripts were identified using complete genomic sequences retrieved from Genbank and miRNAs cataloged in miRbase. The results of this alignment between the seed region of the miRNAs with the B19V complete genome identified 1517 miRNAs that showed 100% identity, of which 412 are bound to NS1, VP1, and VP2 transcripts. Based on the number of total binds to the genome, these miRNAs were ranked, and the top five, miR-4799-5p, miR-5690, miR-335-3p, miR-193b-5p, and miR-6771-3p, were selected to evaluate the target genes and signaling pathways in which they act. We identified 214 common genes among the top five miRNAs, and five of these genes bind to at least two of these miRNAs. Based on WikiPathways and KEGG, these 214 genes act on 29 statistically significant pathways, and the three main pathways were selected. Our results revealed some miRNAs that may be involved in regulating B19V replication and that can act as potential biomarkers for the prognosis of infection.

## 1. Introduction

Parvovirus B19 (B19V) belongs to the *Erythroparvovirus* genus in the *Parvoviridae* family [1]. It is a pathogenic virus widely spread in the human population and is typically transmitted through the respiratory route. However, it can also be transmitted through blood transfusion and to the fetus. Infection happens in erythroid progenitors, fetal cardiac myocytes, and placental endothelial cells [2,3,4,5,6,7].

The B19V infection can lead to a wide range of pathologies and clinical symptoms, which depend on the physiological and immune status of the infected individuals. Common manifestations of B19V infection include erythema infectiosum in children or post-infection and arthropathies mainly in adults [8]. The infection has been linked to severe conditions such as transient aplastic crisis associated with thrombocytopenia, neutropenia, or pancytopenia in individuals with hemolytic disorder; persistent anemia in immunocompromised patients; and hydrops fetalis or miscarriage in cases of intrauterine transmission [9,10,11].

Following primary infection, the B19V DNA can remain detectable in human tissues, in both symptomatic and asymptomatic individuals, particularly in the bone marrow, liver, heart, and synovia [12]. The virus’s ability to persist in tissues has generated significant interest in its potential role in causing inflammatory and chronic diseases, such as myocarditis, hepatitis, and glomerulonephritis [13]. However, the prolonged presence of the viral genome in myocardium, liver, and glomeruli, as well as the cellular and molecular mechanisms involved in nonpermissive cells, are poorly understood.

The virus has a remarkable tropism for human erythroid progenitor cells (EPCs). In addition to the P antigen (globoside), a cellular receptor required for virus entry into cells, an α5β1 integrin complex, and Ku80 are also necessary as co-receptors for virus entry into cells [14,15,16]. However, the antigen P and these co-receptors are expressed in nonpermissive cells [17,18]. Hence, other mechanisms may play a role in the resistance of nonpermissive cells to B19V infection.

After virus entry into permissive cells, ssDNA is converted into a double-stranded replicative form (dsDNA) transcribed in a single precursor mRNA. This precursor mRNA undergoes alternative splicing and polyadenylation. It transcribes 12 B19V mRNAs that encode two structural proteins (VP1 and VP2) that form the viral capsid and the nonstructural protein NS1, which plays a vital role in the replication and transcription of the viral genome [19,20,21]. The transcripts R1 and R1’ translate the nonstructural protein NS1 [22], R4 and R5 mRNAs translate the VP1 structural protein, and R6 and R7 translate the VP2 structural protein [23]. The R8 and R9 mRNAs transcribe an 11 kDa nonstructural protein, which participates in cell signaling and acts as an apoptosis inducer [20,24]. Nonstructural protein 1 (NS1) is produced in permissive and nonpermissive cells. Still, previous studies showed that VP1 and VP2 mRNAs were detected in nonpermissive cells but could not translate to proteins [25,26]. Hence, it has been proposed that there is a possible mechanism for inhibiting the translation of VP mRNAs in B19V nonpermissive cells. Berillo et al. [27] and Anbarlou et al. [28] showed the role of microRNAs (miRNAs) in the inhibition of VP mRNA expression in nonpermissive cells. However, further studies are required to demonstrate which miRNAs regulate B19V replication [27,28].

MicroRNAs are post-transcriptional regulatory noncoding RNAs involved in various biological processes such as cell differentiation, proliferation, and survival. They bind to complementary target messenger RNAs (mRNAs), leading to either translational inhibition or degradation [29]. miRNA-binding sites in viral genomes are primarily located in the non-translated regions (NTRs) but can also be found in the coding regions of viral proteins. There is a complex network of interactions between infected host cells and viruses in which miRNAs can play a crucial role, changing several biological processes, including antiviral defense [30,31]. Viral genomes and gene transcripts exploit passive mechanisms to deregulate host miRNA activity, such as mimicking or blocking the binding between a host miRNA and its target transcript, a phenomenon mediated by the miRNA seed site at the 5’ end of miRNA [32].

We conducted a bioinformatics analysis to predict human miRNAs that may bind to B19V transcripts, specifically the NS1, VP1, and VP2 proteins. Our goal was to identify miRNAs that could regulate B19V replication by targeting related genes and signaling pathways. The structural proteins VP1 and VP2 are significant for B19V replication and immune response induction. Meanwhile, the NS1 protein is a crucial multifunctional protein of parvovirus, exhibiting binding activity to specific DNA sites and functioning as an ATPase, endonuclease, and helicase [33,34,35,36]. Moreover, NS1 is closely associated with host cell apoptosis [37], cell cycle arrest [38], and tissue damage during viral infection [39], making it essential to the pathogenesis of B19V. Additionally, previous reports have indicated that in cells non-permissive to B19V, miRNAs can target the 3’ UTR region of VP mRNAs to inhibit the translation of capsid proteins [27,28]. These data can contribute to a better understanding of the roles of miRNAs in parvovirus-host interactions and highlight potential targets during infection, driving new therapeutic approaches through targeted inhibition of specific cellular metabolic pathways.

## 2. Results

### 2.1. Alignment of miRNAs to the B19V Genome

A total of 19 genomic sequences of Parvovirus B19 were found in the GenBank database (searched in June 2024) but just 12 of them were complete (with 5596pb), so they were selected to be used in this study (http://www.ncbi.nlm.nih.gov/genbank/; accessed on 15 June 2024). Based on the 12 transcripts produced by the B19V genome, we obtained 144 transcript sequences (12 from each of the 12 B19V genomic sequences). We then performed BLAST (https://ftp.ncbi.nlm.nih.gov/blast/executables/blast+/LATEST/; accessed on 15 June 2024) analysis with the 2656 miRNAs available in miRBase. Using the miRBase database (http://www.mirbase.org; version 22.1; accessed on 15 June 2024), we identified 1517 mature microRNAs that perfectly aligned the seed region (100% identity) with the 144 transcripts. The miRNA binding regions were distributed throughout the genome, and the highest number of miRNA binding sites was observed in the NS1 and VP1-unique regions, as shown in Figure 1.

### 2.2. Selection of the miRNAs

After alignment, out of 1517 miRNAs, 1105 were excluded as they did not bind to NS1, VP1, and VP2 transcripts. The remaining 412 miRNAs were ranked based on the number of total bindings to NS1, VP1, and VP2 transcripts, and then we identified the top five miRNAs (Figure 2 and Table 1). After finding potential miRNAs, we identified their targets using three databases to understand the molecular mechanisms underlying the effect on the infection.

### 2.3. Prediction of miRNA Target Genes and the miRNA-mRNA Network

Three databases (miRabel, miRDB, and TargetScan) were searched to find target genes of miR-193b-5p, miR-335-3p, miR-6771-3p, miR-5690, and miR-4799-5p. The duplicates were first removed. The target genes 10, 112, 26, 26, and 40 were identified (totalizing 214 genes) by overlapping analysis for miR-193b-5p, miR-335-3p, miR-6771-3p, miR-5690, and miR-4799-5p, respectively, and the Venn plot was used to show the data (Figure 3).

The miRNA-mRNA network, produced by Cytoscape v.3.10.2, illustrates interactions of miRNAs and intersection genes. Five common genes were shared with at least two miRNAs (*ZNF264*, *ZFP36L1*, *RSL1D1*, *TRDN*, and *SH3KBP1*). The most likely target and common genes shared were *ZNF264* between miR-6771-3p and miR-335-3p; *ZFP36L1* and *RSL1D1* between miR-193b-5p and miR-335-3p; and *TRDN* and *SH3KBP1* between miR-4799-5p and miR-335-3p. In this analysis, the miR-5690 did not share target genes with other miRNAs (Figure 4).

### 2.4. Pathway Enrichment Analysis

The pathways under the influence of all 214 target genes were identified by submitting to the enrichment analyses of EnrichR software, in which 269 and 176 common signaling pathways to all 214 genes were found by WikiPathways and KEGG databases, respectively. Of these, the pathways that presented statistical significance (*p* < 0.05) were selected: 18 from WikiPathways (Figure 5A) and 8 from KEGG (Figure 5B). From them, pathways were ranked according to low *p*-value and bubble size, simultaneously, as these suggest GO terms that are significantly enriched and involve a substantial number of genes. Larger bubbles signify a higher number of genes associated with a particular process or pathway. The cancer-related pathways were excluded (*n* = 6). Figure 5A illustrates the main WikiPathways enriched pathways: miRNA regulation of DNA damage response and DNA damage response. Another important enriched KEGG pathway, retrieved from EnrichR, was the Herpes simplex 1 virus pathway (Figure 5B).

## 3. Discussion

MicroRNAs (miRNAs) are key regulators of gene expression and play significant roles in host–virus interactions [40,41]. Depending on the context, they can either promote or inhibit viral infections, and previous studies suggested that viral genomes may have evolved to directly interact with host miRNAs to facilitate specific steps of replication and progression [41,42,43]. Because of that, viruses may gain an advantage by re-shaping the cellular miRNA availability [44,45,46,47].

To explore their involvement in B19V infection, we identified miRNAs that may influence disease progression and contribute to heterogeneous outcomes. This could help uncover new potential biomarkers for diagnosing and monitoring B19V-related diseases and therapeutic strategies to modulate immune responses, control viral replication, and mitigate tissue damage associated with B19V infection. We also analyzed associated signaling pathways to better understand B19V mechanisms of pathogenesis that lead to divergent outcomes in different populational groups, for instance, regulating cell cycle arrest that B19V exploits [48], modulating immune response, and contributing to clinical manifestations by altering inflammatory pathways or hematopoiesis [49].

Two previous studies evaluated miRNA binding to the VP region of B19V, yielding different results: Berillo et al. [27] identified two significant miRNAs (miR-548an in all 64 isolates and miR-4500 in 63 isolates) in VP2 mRNA, proposing that miRNAs inhibit the translation of viral capsid proteins in non-permissive cells. Anbarlou et al. [28] found 53 miRNAs that could bind to the VP region, demonstrating that disrupting miRNA biogenesis can alleviate the restriction of viral protein expression in non-permissive cells. Together, these studies showed the effect of miRNAs on VP expression, suggesting that miRNAs are involved in permissiveness to B19V infection in erythroid progenitor cells. However, there was no overlap between the analyses and miRNAs found in this present study, which demonstrated miRNAs with the potential role in replication and immunology of the infection. The miRNAs mentioned probably differ because of different goals, methods, and scopes of the studies, such as differences in target regions (VP vs. NS1), prediction software used that can miss some real miRNAs or predict false ones, and the sequence variability of B19V deposited in Genbank.

Our in silico analysis revealed miRNAs—miR-335-3p, miR-193b-5p, miR-4799-5p, miR-5690, and miR-6771-3p—as potential regulators of B19V transcripts VP1, VP2, and NS1. Among them, miR-335-3p and miR-193b-5p have been previously studied in the context of viral infections [50,51,52,53,54], whereas miR-4799-5p, miR-5690, and miR-6771-3p are newly identified targets. This highlights the need to investigate these novel miRNAs further to understand their roles in B19V infection and pathogenesis.

Previous studies reported an association of the miR-335-3p with other viral infections such as SARS-CoV-2 and Epstein Barr. According to Srivastava et al. [55], miR-335-3p was upregulated in the peripheral blood of severe patients infected with SARS-CoV-2 compared to the healthy control group, suggesting this as a potential biomarker to predict disease severity. An in silico study about Epstein Barr virus (EBV) associated with gastric cancer identified miR-335-3p as one of the top 5 miRNAs involved in the EBV-related miRNA regulatory network [56].

miR-193b-5p also played a role in viral infection, being upregulated in Influenza infections, HIV-1 mono-infection, and HIV-HCV co-infection, as well as in cirrhosis and HBV-associated hepatocellular cancer, and downregulated in patients with Japanese encephalitis virus and with HBV-associated acute liver failure. Vaswani and collaborators [50] showed that miR-193b-5p is increased in patients with severe acute respiratory syndrome caused by Influenza virus infection, and the inhibition of this miRNA causes an increase in interferon and genes regulated by it, which leads to a reduction in viral replication, histological evidence of lung damage, and mortality. These studies showed the relevance of these miRNAs in viral infections, mainly in respiratory transmitted infections, such as B19V, SARS-CoV-2, and Influenza.

In serum samples from individuals with cirrhosis and hepatocellular cancer related to hepatitis B virus, miR-193b-5p was found to be upregulated [51]. Franco et al. [52] demonstrated that in patients living with HIV, this miRNA is upregulated in those solely infected with HIV-1, as well as in individuals co-infected with HCV. This upregulation is associated with a significant increase in liver transaminases and shows a strong correlation with the progression of liver fibrosis. In patients with the Japanese encephalitis virus, a study showed that miR-193b-5p was downregulated, thereby upregulating genes involved in interferon signaling pathways, viral genome replication, and defense regulation in response to viruses [53]. In patients with acute liver failure associated with the hepatitis B virus, this miRNA was also negatively regulated [54]. Since previous studies have demonstrated B19V association with acute liver failure in transplanted patients [57], these findings suggest that this miRNA could also have a common role in B19V-associated hepatic injury.

The databases revealed the most relevant pathways with statistical significance: DNA damage response (DDR) and the Herpes simplex 1 virus pathway. The DNA damage response (DDR) is a highly conserved mechanism within the cells to resist DNA damage induced by external and internal factors, such as integration of viral genome, DNA mismatch, or the influence of environmental physical and chemical factors [58]. Multiple DNA viruses have developed relevant strategies to fight host DDR or use host DDR to complete their life cycle [59,60,61]. Previous studies have shown that DNA damage signaling is required for parvovirus replication [62,63], and NS1 plays an essential role in this process. B19V infection induced by the host cell DDR is critical to viral DNA replication by activating all three PI3K kinases (phosphatidylinositol-3 kinase-related kinase) [64]. Recruitment of DDR kinases by viruses at the sites of DNA damage leads to cyclin-dependent kinases (CDKs) silencing and cell cycle arrest [65]. The NS1 of B19V leads to arrested cells by inducing the phosphorylation levels of Cyclin A, Cyclin B1, and cell division Cyclin 2 (CDC2) during viral replication, resulting in up-regulation of CDC2-cyclin B1 complex kinase activity [66]. This results in G2/M checkpoint stasis of the cell cycle, which induces cysteinyl aspartate-specific proteinase (caspases) activation and DNA fragmentation, thus triggering apoptosis [67]. These studies indicate that Parvovirus has evolved a mechanism using the DDR pathway to support virus replication, and the DDR induced by viral infection is of great significance in viral replication and pathogenesis [64].

The other pathway considered relevant was the Herpes simplex 1 virus (HSV-1) infection pathway. Although HSV-1 and B19V are distinct viruses with different structures, target cells, and primary disease manifestations, there are some ways their infection pathways and immune interactions might intersect or relate, particularly in how they evade innate immunity, suppress antiviral signaling, and ensure persistence. Both viruses can suppress type I IFN signaling to implement the immune evasion strategy, facilitating the establishment of virus infection status: HSV-ICP34.5 blocks STAT1/STAT2 activation and dephosphorylates eIF2α to resist the PKR pathway and HSV-ICP0, ICP27, and UL41 suppress IFN-β expression by interfering with IRF3/7 [68]. B19V downregulates IFN-β and IRF signaling in infected erythroid cells by NS1 [69]. Additionally, both viruses manipulate apoptosis and DDR to regulate host cell fate—HSV-1 encodes anti-apoptotic proteins (e.g., Us3) to prevent early cell death, while B19V NS1 induces apoptosis in erythroid cells (pro-virus strategy to assist in viral release) [70]. HSV-1 hijacks DDR pathways to support its replication, while B19V also manipulates DDR (via NS1) to arrest the cell cycle and promote viral replication [48]. These studies underscore related pathways employed by B19V and HSV-1 in manipulating host cell fate through apoptosis and the DNA damage response and immune modulation, reflecting their related approaches to replication and persistence.

This study has significant limitations that are inherent to in silico analyses, such as the reliance on public databases; computational tools can predict miRNAs that bind to a target mRNA even when it does not happen in real biological systems (false positive results). Some true interactions might be missed because many algorithms rely mainly on sequence complementarity and do not catch all real biological interactions (false negative results). Computational results alone do not show whether miRNA binding changes protein production or mRNA stability—this must be tested by experimental studies in a bench laboratory [71,72,73].

As the catalog of miRNA bioinformatics resources continues to expand, ranging from genome-wide target predictors to context-aware functional analyzers, researchers often encounter divergent outputs when using different platforms. This highlights the importance of selecting tools that align with specific experimental goals. For instance, MIRZA-G employs a biophysical thermodynamic framework to predict both canonical and non-canonical binding sites with high accuracy, outperforming many existing programs in off-target detection. Meanwhile, TarPmiR utilizes a random-forest classifier trained on CLASH-derived interaction features, achieving over 74% true-positive recall in human and mouse datasets by integrating seven novel predictive features alongside conventional metrics.

Complementarily, miTALOS v2 integrates high-quality targeting data, tissue-specific gene expression profiles, and pathway annotations to elucidate context-dependent miRNA functions across biological systems. In parallel, MiRmap synthesizes thermodynamic, evolutionary, probabilistic, and sequence-based metrics into an 11-feature repression-strength ranking, providing a comprehensive assessment of silencing potential.

Despite the robust predictive power of these in silico methodologies, downstream bench validation—through reporter assays, targeted expression analyses, and functional in vivo studies—remains essential to substantiate computational predictions and mitigate the risk of misinterpretation due to tool-specific biases.

Experimental studies are needed to validate our findings. As a general approach, the authenticity of the preferred functional miRNA-mRNA target pair, predicted by computational tools, should be validated in the biological model of interest, co-expression of miRNA, and predicted target mRNA. The co-profiling of miRNAs and mRNAs can allow a direct assessment of whether mRNAs are in part shaped by regulatory miRNAs [74] since co-expressed elements share the same transcriptional program or are regulated by members of the same pathway. Consequently, if they are not co-expressed or co-expression cannot be verified in cells or tissues, there is no reason to move forward with additional experiments. Microarray profiling and the latest RNA-sequencing (RNA-seq) represent powerful strategies to carry out large-scale studies on the genome. Moreover, Northern blots and quantitative real-time PCR (RT-qPCR), employing nucleic acids extracted from different cell cultures, represent the most exploited methods to demonstrate co-expression [75,76]. To demonstrate miRNA-mediated effects on target protein expression, protein changes can be detected by conventional procedures such as Western blotting, ELISA, and immunocytochemistry experiments [77].

## 4. Materials and Methods

### 4.1. Identification of miRNAs That Bind to the B19V NS1, VP1, and VP2 Transcripts

To identify miRNAs that bind to B19V NS1, VP1, and VP2 transcripts that are the main proteins involved in viral pathogenesis, we accessed the complete genomic sequences of B19V from the GenBank database of NCBI (https://www.ncbi.nlm.nih.gov/genbank/; accessed on 15 June 2024), since many GenBank entries contained only partial and non-consensus sequences of NS1, VP1, and VP2. During data retrieval, we identified sixteen whole genome sequences by accession numbers NC_000883, MT988400, MT988402, MT682520, MT410187, MT410189, MH201455, MH201456, KM393163, KM393164, KM393165, KM393166, KM393167, KM393168, KM393169, and AY386330. However, four of these sequences showed extensive ambiguous regions denoted by “N” bases (MT988400, MT988402, MT682520, MT410189), indicative of low sequencing coverage or assembly uncertainty. Therefore, we used the twelve remaining full sequences in this study.

Currently, the miRbase v.22.1 (https://www.mirbase.org; accessed on 15 June 2024) identifies 2656 mature sequences of microRNAs in the human genome. The bedtools v2.31.0 getfasta program (https://bedtools.readthedocs.io/en/latest/content/tools/getfasta.html; accessed on 15 June 2024) was used to extract the seed regions of the miRNAs.

miRNA-binding sites in viral genomes are primarily located in the non-translated regions (NTRs) but they can also be found in the coding regions of viral proteins [78]. Since B19V has a single-stranded DNA genome, cellular miRNAs cannot bind directly to the viral genome. Therefore, we looked for miRNAs potentially able to bind to all B19V transcripts (mRNA) (R1 and R1’; R2 and R2’, R3 and R3’, R4 and R5, R6 and R7, R8 and R9) [18]. To find miRNA sites along mRNAs, the transcripts were extracted from the genome coordinates in the sequences retrieved from Genbank (NC_000883) [79]. Subsequently, to search for microRNAs that interact with B19V genome transcripts, we used the command line BLAST algorithm v.2.16.0 (https://ftp.ncbi.nlm.nih.gov/blast/executables/blast+/LATEST/; accessed on 15 June 2024), requiring a perfect alignment (100% identity) of 7 nucleotides, which covered the seed region. Of all the identified miRNAs, those that bound to transcripts of NS1 (R1 and R1’), VP1 (R4 and R5), and VP2 (R6 and R7), from all recovered genomes, were selected and then ranked according to the total number of binds to the B19V genome. This criterion was carefully chosen because the more a miRNA interacts with the genome, the more likely its role in replication and/or antiviral response is significant.

### 4.2. Prediction of miRNA Targets

To predict the potential target mRNAs of selected miRNAs, we combined the predicted target genes from three databases: MiRabel v.2.0 [80], MirDB v.6.0 [81], and TargetScan v.8.0 [82]. The MiRabel tool covers experimentally validated miRNA targets and combines the results from other prediction algorithms (miRanda v.3.3a [83], PITA [84], SVMicrO [85], and TargetScan v.8.0 [82]) that use different and complementary features with interactions, such as seed match, free energy, site accessibility, and target-site abundance, as well as from miRwalk v.2.0 database [86]. The duplicates were eliminated, and then the Venn plot was applied to analyze overlapping genes. A miRNA-mRNA target network was built in Cytoscape 3.10.2 (https://cytoscape.org; accessed on 18 January 2025). This analysis identified target genes that were shared by at least two miRNAs.

### 4.3. Pathway Enrichment Analysis of miRNAs

From the common target genes to all miRNAs, a search for common signaling pathways to the target genes was performed in EnrichR [87], which is a web-server search engine that contains hundreds of thousands of annotated gene sets, from which the pathways from the KEGG and WikiPathways databases were extracted. Of all the pathways found, those that presented statistical significance (*p* < 0.05) were selected, and cancer-related pathways were excluded. The main pathways were selected for analysis based on high gene ratio and low *p*-value, simultaneously (Figure 6).

## 5. Conclusions

Our approach has identified key miRNAs that are potentially associated with B19V targets and provides insights into their association with cellular immune responses and apoptosis pathways, suggesting their role in B19V pathogenesis. However, further studies are necessary to validate these roles experimentally in samples and explore the potential of miRNAs to act as promising biomarkers of infection progression. Understanding the complex interactions between miRNAs and the virus can provide new insights into their use as therapeutic targets for developing antiviral therapies and vaccines.

## Figures and Tables

**Figure 1 ijms-26-05038-f001:**
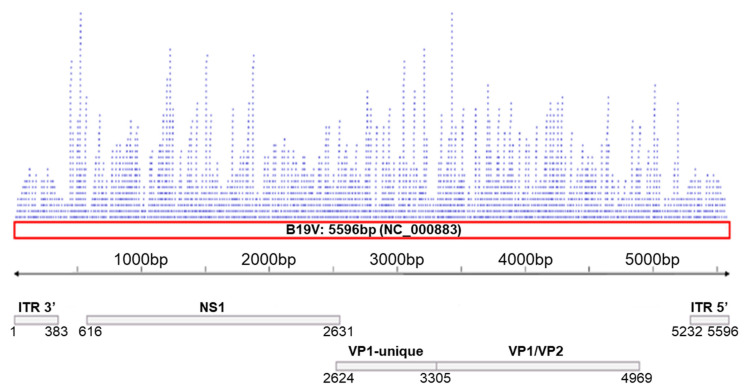
Distribution of the binding sites of the 1517 miRNAs along the B19V genome (NC_000883; 5596 bp; red box).

**Figure 2 ijms-26-05038-f002:**
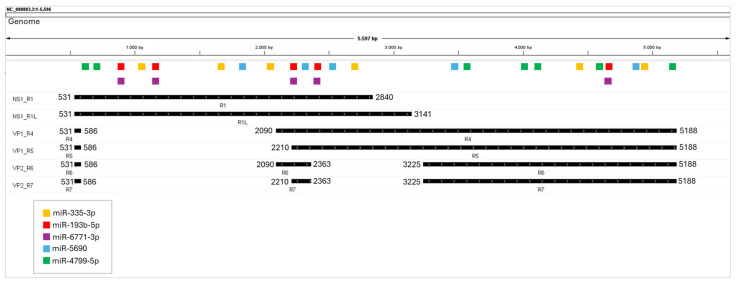
Binding sites of the miRNAs that bind to NS1, VP1, and VP2 transcripts.

**Figure 3 ijms-26-05038-f003:**
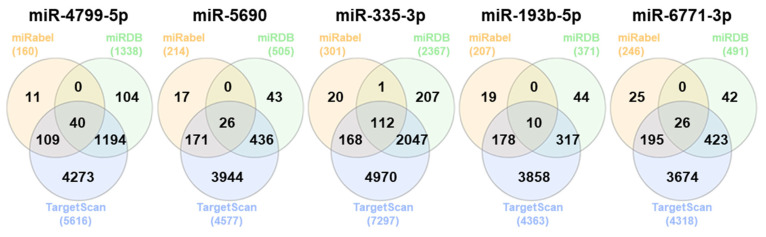
Venn diagrams indicate the number of common target genes for 193b-5p, miR-335-3p, miR-6771-3p, miR-5690, and miR-4799-5p in the three databases (miRabel—yellow, miRDB—green, and TargetScan—blue).

**Figure 4 ijms-26-05038-f004:**
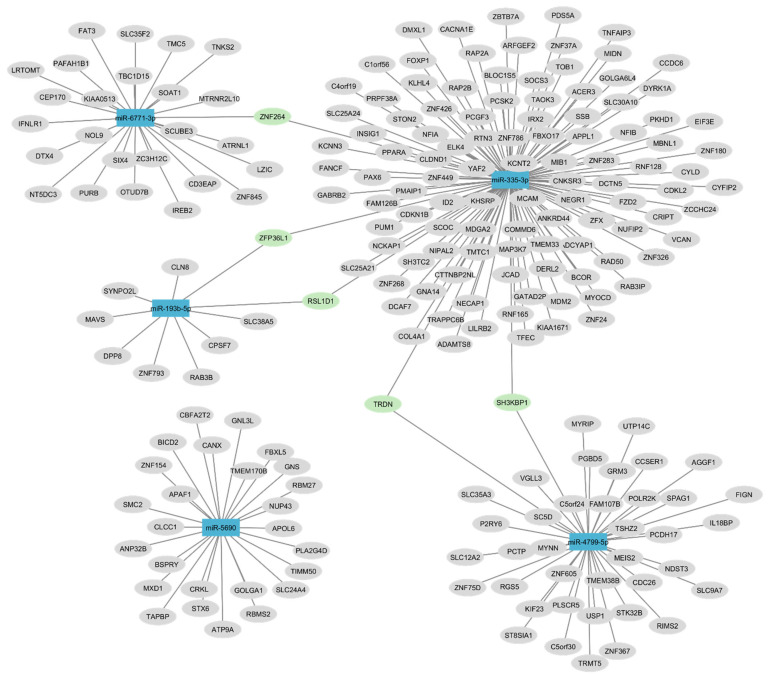
Network maps of 193b-5p, miR-335-3p, miR-6771-3p, miR-5690, and miR-4799-5p and their most common target genes constructed in Cytoscape software v.3.10.2. This analysis identified targets that were shared by two miRNAs. Colors: In blue, the miRNAs are identified; in green, target genes binding at least two miRNAs are identified; and in grey, the other genes are identified.

**Figure 5 ijms-26-05038-f005:**
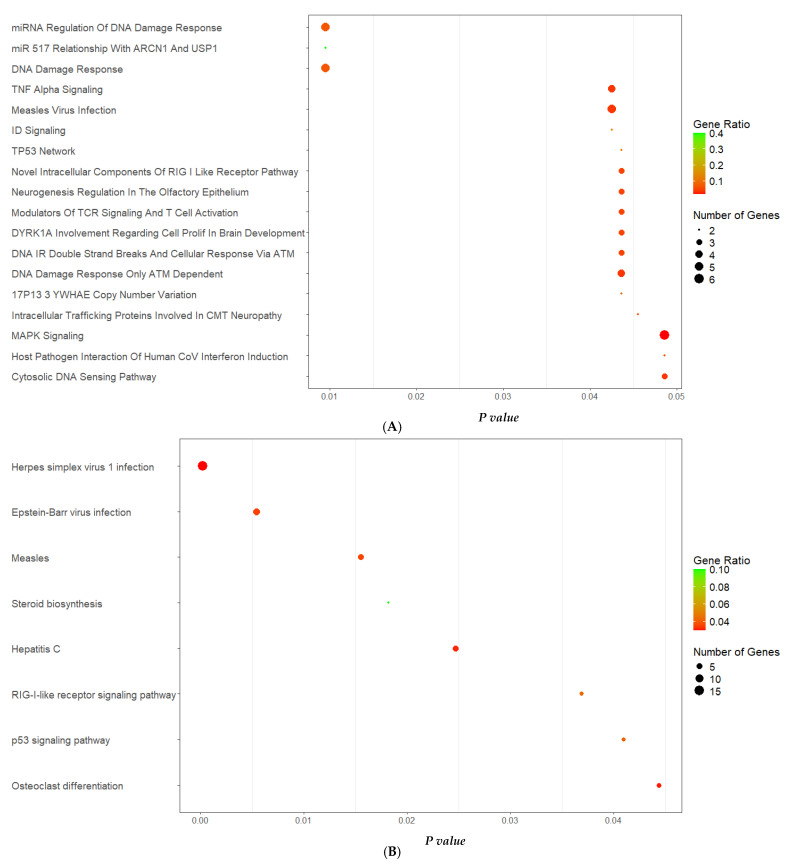
Bubbleplot for statistically significant pathways (*p* < 0.05) from WikiPathways (**A**) and KEGG (**B**) enrichment analysis of 214 common target genes of the top 5 miRNAs. The signaling pathway is assigned to the *y*-axis, and the *p*-value is assigned to the *x*-axis. The area of the circles is proportional to the number of genes assigned, and the color corresponds to the gene ratio (number of input genes/number of all genes in this pathway).

**Figure 6 ijms-26-05038-f006:**
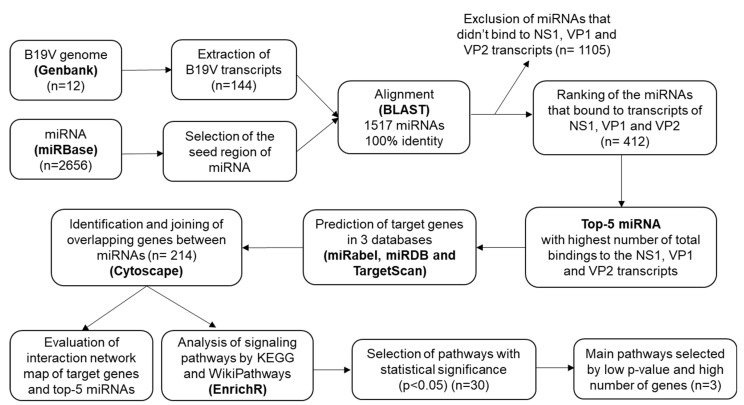
Flowchart of the in silico study.

**Table 1 ijms-26-05038-t001:** Ranking of the miRNAs with the highest interaction with the NS1, VP1, and VP2 B19V transcripts.

Ranking	miRNA	R1	R1’	R4	R5	R6	R7	Score ^1^
1	miR-4799-5p	24	24	56	56	56	56	272
2	miR-5690	35	35	47	47	35	35	234
3	miR-335-3p	48	48	36	36	24	24	216
4	miR-193b-5p	47	47	35	35	24	24	212
5	miR-6771-3p	47	47	35	35	24	24	212

^1^ Score was based on the number of miRNA bindings to the transcript. R1 and R1’ refer to the NS1 transcript; R4 and R5 refer to the VP1 transcript; and R6 and R7 refer to the VP2 transcript.

## Data Availability

All data analyzed in this manuscript were from the following public domain and databases: GenBank database of NCBI (https://www.ncbi.nlm.nih.gov/genbank/, accessed on 5 August 2024); miRbase (https://www.mirbase.org, accessed on 5 August 2024); KEGG (https://www.genome.jp/kegg/, accessed on 5 August 2024); and Wikipathways (https://www.wikipathways.org/, accessed on 5 August 2024).

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
