# Peer review of "Screening out microRNAs and Their Molecular Pathways with a Potential Role in the Regulation of Parvovirus B19 Infection Through In Silico Analysis"

_ijms, 2025, doi:10.3390/ijms26115038_

Round 1

Reviewer 1 Report

Comments and Suggestions for Authors

This study aims to analyze in silico, using bioinformatics resources, the role of miRNAs in the pathogenesis of Parvovirus B19 infections. According to this reviewer, the following points in the manuscript could be re-evaluated:

Lines 46-47: They repeat information from lines 38-39. The text should be reorganized to merge the first and second paragraphs.

Lines 52-53: "Prolonged presence of the viral genome" – where?

Line 66: To what do "R1 and R1'" refer?

Lines 158 and 159: According to Figure 5B, the pathways with the highest statistical significance were Herpes simplex virus 1 infection and Epstein-Barr virus infection, not EBV and measles.

Lines 168-200: They repeat much of the information already presented in the "Results" section; lines 179-182 could be removed, and the rest of the text summarized.

Lines 235-237: According to Figure 5B, the statistically most significant pathways would include Herpes simplex virus, not measles; if Figure 5B is correct, the discussion is incomplete.

Lines 255-275: How are the pieces of information contained in these two paragraphs related to Parvovirus B19 infection?

Lines 276-315: The Materials and Methods section needs to be reorganized. We suggest placing Figure 6 last, after all procedures have been described. The accession numbers of the B19V genomes used should be mentioned. On line 281, the software used must be properly referenced.

Comments on the Quality of English Language

The English needs improvement. There are many repeated words and phrases that are not actually used, likely due to erroneous translations from the original language.

Author Response

Comment: This study aims to analyze in silico, using bioinformatics resources, the role of miRNAs in the pathogenesis of Parvovirus B19 infections. According to this reviewer, the following points in the manuscript could be re-evaluated:
Lines 46-47: They repeat information from lines 38-39. The text should be reorganized to merge the first and second paragraphs.
Reply: Thank you for pointing this out. We removed the sentence “can result in hydrops fetalis and/or fetal death” and reorganized the phrase in lines 38-39.

Comment: Lines 52-53: "Prolonged presence of the viral genome" – where?
Reply: Thank you for pointing this out. We added “in myocardium, liver, and glomeruli” in the phrase.

Comment: Line 66: To what do "R1 and R1'" refer?
Reply: Thank you for pointing this out. We add “The transcripts” in the beginning of the phrase.

Comment: Lines 158 and 159: According to Figure 5B, the pathways with the highest statistical significance were Herpes simplex virus 1 infection and Epstein-Barr virus infection, not EBV and measles.
Reply: Thank you for pointing this out. The main pathways were ranked according to low p-value and bubble size, simultaneously, as these suggest GO terms that are significantly enriched and involve a substantial number of genes. Larger bubbles signify a higher number of genes associated with a particular process or pathway. Therefore, the analysis revealed the most relevant pathways with statistical significance: DNA damage response (DDR) (Fig 5A) and Herpes simplex 1 virus pathway (Fig 5B) (line 163). In the Discussion section, these pathways were better discussed (lines 258-275).

Comment: Lines 168-200: They repeat much of the information already presented in the "Results" section; lines 179-182 could be removed, and the rest of the text summarized.
Reply: Thank you for pointing this out. We removed lines 179-182 and summarized the text from lines 168-200. 

Comment: Lines 235-237: According to Figure 5B, the statistically most significant pathways would include Herpes simplex virus, not measles; if Figure 5B is correct, the discussion is incomplete.
Reply: Thank you for pointing this out. Figure 6 was changed for better understanding. As mentioned above, the main pathways were ranked according to low p-value and bubble size, simultaneously, as these suggest GO terms that are significantly enriched and involve a substantial number of genes. Larger bubbles signify a higher number of genes associated with a particular process or pathway. Therefore, the analysis revealed the most relevant pathways with statistical significance: DNA damage response (DDR) (Fig 5A) and Herpes simplex 1 virus pathway (Fig 5B) (line 163). In the Discussion section, these pathways were better discussed (lines 258-275).

Comment: Lines 255-275: How are the pieces of information contained in these two paragraphs related to Parvovirus B19 infection?
Reply: Thank you for pointing this out. Figure 6 was changed for better understanding, and the Herpes simplex 1 virus (HSV-1) infection was considered more relevant (Figure 5B). In the Discussion section, it was discussed the importance of this pathway and its possible relationship with parvovirus B19 infection (lines 258-275).  

Comment: Lines 276-315: The Materials and Methods section needs to be reorganized. We suggest placing Figure 6 last, after all procedures have been described. The accession numbers of the B19V genomes used should be mentioned. On line 281, the software used must be properly referenced.
Reply: Thank you for pointing this out. We moved Figure 6 to the end of the topic and added the accession numbers of B19V genomes (NC_000883, MT988400, MT988402, MT682520, MT410187, MT410189, MH201455, MH201456, KM393163, KM393164, KM393165, KM393166, KM393167, KM393168, KM393169 and AY386330) on lines 266-269. The software reference was added in lines 274-275 as well. 

Reviewer 2 Report

Comments and Suggestions for Authors

This manuscript aims to investigate the role of microRNAs (miRNAs) in regulating Parvovirus B19 (B19V) infection through in silico analysis. The study identifies potential miRNAs that bind to B19V transcripts and predicts their target genes and signaling pathways. The topic is novel and holds significant scientific value, particularly given the complex interactions between viruses and host miRNAs.

Suggestions for Improvement

Strengthen the Discussion‌:

Expand on the potential implications of the identified miRNAs for B19V pathogenesis and their potential as biomarkers or therapeutic targets.
Compare and contrast the current findings with previous studies on miRNA-virus interactions, discussing consistencies and discrepancies.
Discuss the limitations of in silico analyses and propose experimental validation strategies.

Enhance Experimental Validation‌:

Acknowledge the need for experimental validation of the predicted miRNA-B19V interactions and target genes.
Suggest potential experimental approaches, such as qPCR, Western blotting, or luciferase reporter assays, to verify the findings.

Clarify Technical Details‌:

Provide more details on the bioinformatics tools and databases used, including versions and parameters.
Explain the criteria for selecting the top miRNAs and the rationale behind the cut-off values.

Improve Figure Quality‌:

Enhance the resolution and clarity of the figures, particularly Figure 1 and Figure 5, to improve readability.
Consider adding labels or annotations to the figures to guide the reader.

Expand on Limitations‌:

Discuss the limitations of the study, such as the reliance on public databases, potential biases in the data, and the lack of experimental validation.
Suggest directions for future research to address these limitations.

Minor Comments
Consistency in Abbreviations‌: Ensure consistency in the use of abbreviations, particularly for databases and tools.
Formatting‌: Adhere to the journal's formatting guidelines, including font size, line spacing, and citation style.

Author Response

Comment: This manuscript aims to investigate the role of microRNAs (miRNAs) in regulating Parvovirus B19 (B19V) infection through in silico analysis. The study identifies potential miRNAs that bind to B19V transcripts and predicts their target genes and signaling pathways. The topic is novel and holds significant scientific value, particularly given the complex interactions between viruses and host miRNAs.
Suggestions for Improvement
Strengthen the Discussion:
Expand on the potential implications of the identified miRNAs for B19V pathogenesis and their potential as biomarkers or therapeutic targets.
Reply: Thank you for pointing this out. The discussion was improved as recommended by the reviewer. 
Lines 183-189: “This could help uncover new potential biomarkers for diagnosing and monitoring B19V-related diseases and therapeutic strategies to modulate immune responses, control viral replication, and mitigate tissue damage associated with B19V infection. We also analyzed associated signaling pathways to better understand mechanisms of pathogenesis that lead to divergent outcomes in different populational groups, for instance, regulating cell cycle arrest that B19V exploits, modulating immune response and contributing to clinical manifestations by altering inflammatory pathway or hematopoiesis.

 Comment: Compare and contrast the current findings with previous studies on miRNA-virus interactions, discussing consistencies and discrepancies.
Reply: Thank you for pointing this out. There are only 2 studies on miRNA and Parvovirus B19 interaction. The discussion between them was improved in our manuscript in lines 190-203.

Comment: Discuss the limitations of in silico analyses and propose experimental validation strategies.
Reply: Thank you for pointing this out. We improve our discussion by including a paragraph about limitations of the in silico analyses (lines 277-284) and proposing and explaining experimental validation studies (lines 284-298). 

Comment: Enhance Experimental Validation:
Acknowledge the need for experimental validation of the predicted miRNA-B19V interactions and target genes.
Reply: Thank you for pointing this out. As mentioned above, we improved our discussion by including a paragraph proposing and explaining experimental validation studies (lines 284-298) 

Comment: Suggest potential experimental approaches, such as qPCR, Western blotting, or luciferase reporter assays, to verify the findings.
Reply: Thank you for pointing this out. As suggested by the reviewer, we added potential experimental approaches, such as qPCR, Western blotting, or luciferase reporter assays, to verify our findings, in the discussion section (lines 284-298).

Comment: Clarify Technical Details:
Provide more details on the bioinformatics tools and databases used, including versions and parameters.
Reply: Thank you for pointing this out. We have included in the manuscript the specific versions of all tools and databases used (where multiple versions existed) and clarified that all parameters were run with their default settings.

Comment: Explain the criteria for selecting the top miRNAs and the rationale behind the cut-off values.
Reply: Thank you for pointing this out. For better understanding, we added a supplementary table with full results about the ranking score of the miRNAs obtained. 
In our conservative selection of the top five miRNAs, we prioritized those with the highest multiplicity of canonical 7-mer seed sites across the NS1, VP1, and VP2 regions. Consequently, isolated point mutations are unlikely to abolish the cumulative binding potential of these candidates. 
During viral infection, miRNAs with multiple binding sites on viral genomes are more likely to engage and mobilize cellular miRNA machinery, amplifying their functional impact. This criteria for selecting the top miRNAs have also been adopted in several previous studies (Zhao et al., 2022; Song et al., 2010; Huang et al., 2007). 
Regarding the rationale behind the cut-off values, their sites exhibited a clear cut-off point beyond the fifth rank, suggesting a natural threshold for meaningful interactions (empirical observation).
-    Zhao, E., Li, X., You, B., Wang, J., Hou, W., & Wu, Q. (2022). Identification of a Five-miRNA Signature for Diagnosis of Kidney Renal Clear Cell Carcinoma. Frontiers in genetics, 13, 857411. https://doi.org/10.3389/fgene.2022.857411
-    Song L, Liu H, Gao S, Jiang W, Huang W. Cellular microRNAs inhibit replication of the H1N1 influenza A virus in infected cells. J Virol. 2010;84(17):8849-8860. doi:10.1128/JVI.00456-10
-    Huang, J., Wang, F., Argyris, E., Chen, K., Liang, Z., Tian, H., Huang, W., Squires, K., Verlinghieri, G., & Zhang, H. (2007). Cellular microRNAs contribute to HIV-1 latency in resting primary CD4+ T lymphocytes. Nature medicine, 13(10), 1241–1247. https://doi.org/10.1038/nm1639
Comment: Improve Figure Quality:
Enhance the resolution and clarity of the figures, particularly Figure 1 and Figure 5, to improve readability.
Reply: Thank you for pointing this out. The resolution of Figures 1 and 5 was enhanced for better viewing. 

Comment: Expand on Limitations:
Discuss the limitations of the study, such as the reliance on public databases, potential biases in the data, and the lack of experimental validation.
Reply: Thank you for pointing this out. As mentioned above, we improved our discussion by including a paragraph about limitations of the in silico analyses (lines 277-284) and proposing and explaining experimental validation studies (lines 284-298). 

Comment: Suggest directions for future research to address these limitations.
Reply: Thank you for pointing this out. We suggest future research by including a paragraph proposing and explaining experimental validation studies (lines 285-295)

Comment: Minor Comments
Consistency in Abbreviations: Ensure consistency in the use of abbreviations, particularly for databases and tools.
Reply: Thank you for pointing this out. In this version of the article, we checked all the abbreviations used, in order to make the text more consistent.

Comment: Formatting: Adhere to the journal's formatting guidelines, including font size, line spacing, and citation style.
Reply: Thank you for pointing this out. The work was based on the journal's formatting guidelines, which are available at the link: https://www.mdpi.com/journal/ijms/instructions. 

Reviewer 3 Report

Comments and Suggestions for Authors

This paper uses cutting-edge methods to search for miRNAs that interact with Parvovirus B19 (PB19) and predict the metabolic pathways in which they are involved. In particular, the results are made objective by examining the pathway with the highest gene ratio in Figure 6. However, the results submitted by the authors at present are merely a search of existing base sequences using existing tools, and no hypothesis-based verification has been performed. The reviewer believes that the relationship between the Parvovirus B19 gene base sequence mutations and the pathogenicity and tissue specificity of these strains and the miRNAs that the authors focused on should be analyzed.

Items that must be added

  • Since the search for miRNA-interacting regions in the entire genome is conducted, the target PB19 base sequence information is limited. On the other hand, the authors search for miRNAs that interact with NS1, VP1, and VP2 transcripts. If the base sequences that interact with miRNAs are limited to NS1, VP1, and VP2 transcripts, a large amount of base sequence information can be obtained from GenBank. For some of these base sequences, changes in the proliferation site and pathogenicity due to mutations have also been reported in papers. The authors should further investigate the interactions between the NS1, VP1, and VP2 sequences described in these papers and the miRNAs in question, and consider the relationship between miRNAs and pathogenicity and tissue localization.
  • Examples of references
    • Ceccarelli, Giancarlo, et al. "Reassessing the Risk of Severe Parvovirus B19 Infection in the Immunocompetent Population: A Call for Vigilance in the Wake of Resurgence." Viruses 16.9 (2024): 1352.
    • Nowlan, Kirsten, et al. "Parvovirus B19 and Human Herpes Virus 6B and 7 Are Frequently Found DNA Viruses in the Human Thymus But Show No Definitive Link With Myasthenia Gravis." The Journal of Infectious Diseases (2024): jiae600.
  • In the Discussion section, the authors mention the role of miRNAs in cellular immune responses and apoptotic pathways. However, the pathway with the highest gene ratio may not be directly related to these. Discussion should be added about the pathway with the highest gene ratio in Figure 6.

Items for improvement

  • The authors identified miRNAs that bind to NS1, VP1, and VP2 transcripts from among the miRNAs and selected the miRNA with the highest binding ability by ranking them, but this work does not require the use of the full-length PB19 genome. The authors should clearly state why they used the full-length PB19 genome.
  • As for the methods described in previous studies such as Bader, S., & Tuller, T. (2024). Advanced computational predictive models of miRNA-mRNA interaction efficiency. Computational and Structural Biotechnology Journal, 23, 1740–1754. https://doi.org/10.1016/j.csbj.2024.04.015, a comparison with the method adopted by the authors should be included in the Discussion section.
    • MIRZA-G - A method for accurately predicting miRNA targets and off-targets of small interfering RNAs.
    • TarPmiR - A novel approach for miRNA target site prediction.
    • miTALOS - A method for analyzing tissue-specific miRNA function.
    • MiRmap - A method for comprehensive prediction of miRNA target silencing potential.

Author Response

Comment: This paper uses cutting-edge methods to search for miRNAs that interact with Parvovirus B19 (PB19) and predict the metabolic pathways in which they are involved. In particular, the results are made objective by examining the pathway with the highest gene ratio in Figure 6. However, the results submitted by the authors at present are merely a search of existing base sequences using existing tools, and no hypothesis-based verification has been performed. The reviewer believes that the relationship between the Parvovirus B19 gene base sequence mutations and the pathogenicity and tissue specificity of these strains and the miRNAs that the authors focused on should be analyzed.
Reply: We appreciate the reviewer’s comment. However, analyzing the B19V gene base sequence mutations and tissue specificity of the B19V strains is not the proposed objective of this study. We hypothesized that there were human miRNAs that could regulate the B19 infection by post-transcriptional regulation of B19V mainly protein (VP1, VP2 and NS1) and therefore, could have a potential role in the pathogenesis of B19V infection.  To verify this hypothesis, we carried out an in silico analysis using existing computational tools to predict the targets and metabolic pathways, as detailed in the paper. However, as with all computational analysis studies, this study has limitations inherent to in silico analysis studies, that is, mainly the need for a following experimental validation of the findings.

Comment:  Items that must be added
Since the search for miRNA-interacting regions in the entire genome is conducted, the target PB19 base sequence information is limited. On the other hand, the authors search for miRNAs that interact with NS1, VP1, and VP2 transcripts. If the base sequences that interact with miRNAs are limited to NS1, VP1, and VP2 transcripts, a large amount of base sequence information can be obtained from GenBank. For some of these base sequences, changes in the proliferation site and pathogenicity due to mutations have also been reported in papers. The authors should further investigate the interactions between the NS1, VP1, and VP2 sequences described in these papers and the miRNAs in question and consider the relationship between miRNAs and pathogenicity and tissue localization.
•    Examples of references
•    Ceccarelli, Giancarlo, et al. "Reassessing the Risk of Severe Parvovirus B19 Infection in the Immunocompetent Population: A Call for Vigilance in the Wake of Resurgence." Viruses 16.9 (2024): 1352.
•    Nowlan, Kirsten, et al. "Parvovirus B19 and Human Herpes Virus 6B and 7 Are Frequently Found DNA Viruses in the Human Thymus But Show No Definitive Link With Myasthenia Gravis." The Journal of Infectious Diseases (2024): jiae600.
Reply: We appreciate the reviewer’s insightful suggestion regarding sequence variation in NS1, VP1, and VP2 and its potential impact on miRNA-binding landscapes. Our primary objective was to analyze fully assembled B19V genomes. During data retrieval, we observed that many GenBank entries contained only partial sequences of NS1, VP1 and VP2, as well as extensive ambiguous regions denoted by “N” bases, indicative of low sequencing coverage or assembly uncertainty. Such regions would compromise precise seed-based miRNA-binding predictions. To ensure genome-quality integrity in downstream analyses, we filtered out sequences with high proportions of ambiguous nucleotides.
Furthermore, human parvovirus B19 exhibits relatively low evolutionary variability compared to RNA viruses, with substitution rates estimated between 1.0 × 10−4 and 4.0 × 10−4 substitutions per site per year. Subgenotype 1a, for instance, shows a rate of approximately 2.6 × 10−4 substitutions/site/year. Phylogenetic analyses of ancient and modern isolates reveal long-term lineage stability spanning millennia. Given this modest mutation rate and overall genomic conservation, we anticipate only sporadic point changes rather than widespread loss of 7-mer seed matches.
In our conservative selection of the top five miRNAs, we prioritized those with the highest multiplicity of canonical 7-mer seed sites across the NS1, VP1, and VP2 regions. Consequently, isolated point mutations are unlikely to abolish the cumulative binding potential of these candidates. We have added this rationale to the Discussion section and emphasize that in silico predictions must be complemented by experimental validation—such as luciferase reporter assays, qPCR quantification, and in vivo infection models—to confirm the functional impact of miRNA binding across viral variants

Comment: In the Discussion section, the authors mention the role of miRNAs in cellular immune responses and apoptotic pathways. However, the pathway with the highest gene ratio may not be directly related to these. Discussion should be added about the pathway with the highest gene ratio in Figure 6.
Reply: Thanks for the comment. Figure 6 was changed for better understanding. The pathways were ranked according to low p-value and bubble size, simultaneously, as these suggest GO terms that are significantly enriched and involve a substantial number of genes. Larger bubbles signify a higher number of genes associated with a particular process or pathway. Therefore, the analysis revealed the most relevant pathways with statistical significance: DNA damage response (DDR) and Herpes simplex 1 virus pathway. In the Discussion section these pathways were better discussed, including the possible role of the miRNAs identified in the study, in the cellular immune responses and apoptotic pathways (lines 240-276). 

Comment: Items for improvement
The authors identified miRNAs that bind to NS1, VP1, and VP2 transcripts from among the miRNAs and selected the miRNA with the highest binding ability by ranking them, but this work does not require the use of the full-length PB19 genome. The authors should clearly state why they used the full-length PB19 genome.
Reply: As mentioned above, our primary objective was to analyze fully assembled B19V genomes. During data retrieval, we observed that many GenBank entries contained only partial sequences of NS1, VP1 and VP2, as well as extensive ambiguous regions denoted by “N” bases, indicative of low sequencing coverage or assembly uncertainty. Such regions would compromise precise seed-based miRNA-binding predictions. To ensure genome-quality integrity in downstream analyses, we filtered out sequences with high proportions of ambiguous nucleotides.  

Comment: As for the methods described in previous studies such as Bader, S., & Tuller, T. (2024). Advanced computational predictive models of miRNA-mRNA interaction efficiency. Computational and Structural Biotechnology Journal, 23, 1740–1754. https://doi.org/10.1016/j.csbj.2024.04.015, a comparison with the method adopted by the authors should be included in the Discussion section.
•    MIRZA-G - A method for accurately predicting miRNA targets and off-targets of small interfering RNAs.
•    TarPmiR - A novel approach for miRNA target site prediction.
•    miTALOS - A method for analyzing tissue-specific miRNA function.
•    MiRmap - A method for comprehensive prediction of miRNA target silencing potential.
Reply: Thank you for pointing this out. As suggested, we included the methods described by the authors cited above, in the Discussion section (lines 285-304).

Round 2

Reviewer 1 Report

Comments and Suggestions for Authors

Regarding the manuscript "Screening out microRNAs and their molecular pathways with a potential role in the regulation of Parvovirus B19 infection  through in silico analysis", it is surprising to this reviewer that, amid such sophisticated analyses, a fundamental error in data interpretation has occurred — specifically, the misinterpretation of Figure 5B, which clearly highlights "herpes simplex infection." While the revisions suggested in the previous round were addressed, it is crucial to emphasize that a basic aspect, such as the correct and careful interpretation of a graph illustrating the obtained results, is not a minor issue. Likewise, a comprehensive discussion of all research approaches is equally important. Due to the initial misinterpretation, a significant portion of the discussion had to be modified in the second version of this manuscript, which may lead to questions about the accuracy and relevance of the topics that were changed. Additionally, we suggest removing paragraphs from lines 285 to 318, as these sections extend beyond the scope of the research presented. 

Author Response

Comment: Regarding the manuscript "Screening out microRNAs and their molecular pathways with a potential role in the regulation of Parvovirus B19 infection through in silico analysis", it is surprising to this reviewer that, amid such sophisticated analyses, a fundamental error in data interpretation has occurred — specifically, the misinterpretation of Figure 5B, which clearly highlights "herpes simplex infection." While the revisions suggested in the previous round were addressed, it is crucial to emphasize that a basic aspect, such as the correct and careful interpretation of a graph illustrating the obtained results, is not a minor issue. Likewise, a comprehensive discussion of all research approaches is equally important. Due to the initial misinterpretation, a significant portion of the discussion had to be modified in the second version of this manuscript, which may lead to questions about the accuracy and relevance of the topics that were changed. Additionally, we suggest removing paragraphs from lines 285 to 318, as these sections extend beyond the scope of the research presented. 

Reply: Thank you for pointing this out. A significant portion of the discussion had to be modified to attend to the requests of all reviewers, including the addition of a comparison with the method adopted by other authors, such as MIRZA-G, TarPmiR, miTALOS, MiRmap (lines 285-303). Another request was to add further potential experimental approaches, such as qPCR, Western blotting, or luciferase reporter assays, to verify the findings of the in silico analysis (lines 304-318). Therefore, we do not feel comfortable removing them without the agreement of the other reviewers.

Reviewer 3 Report

Comments and Suggestions for Authors

Comments on the revised manuscript

Response to comment 1 and in revising the text, the authors state that "We hypothesized that there were human miRNAs that could regulate the B19 infection by post-transcriptional regulation of B19V mainly protein (VP1, VP2 and NS1) and therefore, could have a potential role in the pathogenesis of B19V infection. To verify this hypothesis, we carried out an in silico analysis using existing computational tools to predict the targets and metabolic pathways, as detailed in the paper." and that their goal was to show that there are miRNAs that may interact with the B19V sequence. However, the selection made by the authors was primarily to show the top miRNAs in the ranking of possible interactions. To clearly show the authors' goal, they should not only rank the miRNAs, but also emphasize that they found candidates for interactions that exceed the predicted threshold, and then show the ranking. 

Response to comment 2 and text revision: The authors state, "Our primary objective was to analyze fully assembled B19V genomes. During data retrieval, we observed that many GenBank entries contained only partial sequences of NS1, VP1 and VP2, as well as extensive ambiguous regions denoted by “N” bases, indicative of low sequencing coverage or assembly uncertainty." However, authors did not refer to GenBank Accession as an example in the manuscript. Alternatively, the authors should emphasize that the reason they referenced V1, V2, and NS1 data from the whole genome data was to obtain highly reliable base sequences, and that useful information will be obtained from partial sequences in the future.

Response to comment 3 and text revision: The reviewer emphasized Figure 6, but the fact that the scope of influence of miRNAs includes immunity and apoptosis has been emphasized and added in other parts of the Discussion, so we judge that this comment has been adequately addressed, including the change to Figure 6.

Author Response

Comment: Response to comment 1 and in revising the text, the authors state that "We hypothesized that there were human miRNAs that could regulate the B19 infection by post-transcriptional regulation of B19V mainly protein (VP1, VP2 and NS1) and therefore, could have a potential role in the pathogenesis of B19V infection. To verify this hypothesis, we carried out an in silico analysis using existing computational tools to predict the targets and metabolic pathways, as detailed in the paper." and that their goal was to show that there are miRNAs that may interact with the B19V sequence. However, the selection made by the authors was primarily to show the top miRNAs in the ranking of possible interactions. To clearly show the authors' goal, they should not only rank the miRNAs, but also emphasize that they found candidates for interactions that exceed the predicted threshold, and then show the ranking.

Reply: We appreciate the reviewer’s comment. For better understanding, we added a supplementary table with full results about the ranking score of the miRNAs obtained.

In our conservative selection of the top five miRNAs, we prioritized those with the highest

number of predicted binding-site locations across all investigated targets NS1, VP1, and VP2 (highest multiplicity of canonical 7-mer seed sites across the NS1, VP1, and VP2 regions), as their site counts exhibited a clear inflection point beyond the fifth rank, suggesting a natural threshold for meaningful interactions.  These criteria for selecting the top miRNAs have also been adopted in several previous studies (Zhao et al., 2022; Song et al., 2010; Huang et al., 2007).

  • Zhao, E., Li, X., You, B., Wang, J., Hou, W., & Wu, Q. (2022). Identification of a Five-miRNA Signature for Diagnosis of Kidney Renal Clear Cell Carcinoma. Frontiers in genetics, 13, 857411. https://doi.org/10.3389/fgene.2022.857411
  • Song L, Liu H, Gao S, Jiang W, Huang W. Cellular microRNAs inhibit replication of the H1N1 influenza A virus in infected cells. J Virol. 2010;84(17):8849-8860. doi:10.1128/JVI.00456-10
  • Huang, J., Wang, F., Argyris, E., Chen, K., Liang, Z., Tian, H., Huang, W., Squires, K., Verlinghieri, G., & Zhang, H. (2007). Cellular microRNAs contribute to HIV-1 latency in resting primary CD4+ T lymphocytes. Nature medicine, 13(10), 1241–1247. https://doi.org/10.1038/nm1639

Comment: Response to comment 2 and text revision: The authors state, "Our primary objective was to analyze fully assembled B19V genomes. During data retrieval, we observed that many GenBank entries contained only partial sequences of NS1, VP1 and VP2, as well as extensive ambiguous regions denoted by “N” bases, indicative of low sequencing coverage or assembly uncertainty." However, authors did not refer to GenBank Accession as an example in the manuscript. Alternatively, the authors should emphasize that the reason they referenced V1, V2, and NS1 data from the whole genome data was to obtain highly reliable base sequences, and that useful information will be obtained from partial sequences in the future.

Reply: We appreciate the reviewer’s comment. For better understanding, we changed the text at lines 320-332. Additionally, the reason for referring to VP1, VP2, and NS1 was due to the importance of these proteins in the pathogenesis, as has been emphasized at lines 90-101 of the manuscript.

Comment: Response to comment 3 and text revision: The reviewer emphasized Figure 6, but the fact that the scope of influence of miRNAs includes immunity and apoptosis has been emphasized and added in other parts of the Discussion, so we judge that this comment has been adequately addressed, including the change to Figure 6.

Reply: Thank you for the comment.